# Sugar–Acetic Acid–Ethanol–Water Mixture as a Potent Attractant for Trapping the Oriental Fruit Moth (Lepidoptera: Tortricidae) in Peach–Apple Mixed-Planting Orchards

**DOI:** 10.3390/plants8100401

**Published:** 2019-10-08

**Authors:** Hao Zhai, Xianmei Yu, Yanan Ma, Yong Zhang, Dan Wang

**Affiliations:** Shandong Institute of Pomology, Tai’an, Shandong 271000, China; zhaihao688@163.com (H.Z.); yuxianmei95@163.com (X.Y.); ynma85@163.com (Y.M.)

**Keywords:** sugar–acetic acid–ethanol–water mixture (SAEWM), oriental fruit moth, trapping efficacy, female/male ratio, natural enemy insects, green pest control

## Abstract

Sugar–acetic acid–ethanol–water mixture (SAEWM) trapping has initially shown the potential efficacy for monitoring or trapping insects. It is unknown how SAEWM-baited traps affect field number of oriental fruit moth (OFM), *Grapholita molesta* (Busck) (Lepidoptera: Tortricidae), the female/male ratio trapped, and the type of natural-enemy insects captured. This study investigated changes in seasonal population dynamics and diurnal flight rhythm of OFM, the number and female/male ratio of OFM and the numbers of *Coccinellidae* and *Chrysopidae* trapped by SAEWM in peach–apple mixed-planting orchards. The SAEWM performed well in trapping OFM, most of which were adult females, with the maximum trapping at 2.5 m above ground. The daily trapping peak occurred between 18:00 and 20:00, during each continuous monitoring period, with another peak occurring at 4:00–8:00, after the second monitoring period (2–5 July). However, the use of SAEWM also resulted in the trapping of *Coccinellidae* and *Chrysopidae*, of which peak trapping time partially overlapped with the second and third peak trapping times of OFM. We suggest the cessation of SAEWM trapping during the peak activity time of *Coccinellidae* and *Chrysopidae*, or application of alternative attractive mixture that do not trap the natural enemy insects, in order to protect the ecological balance in the field.

## 1. Introduction

The oriental fruit moth (OFM), *Grapholita molesta* (Busck) (Lepidoptera: Tortricidae) is a notorious pest present in most of the fruit-producing areas worldwide and can damage the tender shoots and fruits of many fruit trees, including peach, apple, pear, plum, and cherry trees [1,2]. OFM typically switch plant parts and hosts seasonally, primarily damaging shoots in peach orchards from spring to early summer and boring into fruits in apple and pear orchards from late summer to autumn in China, which is a major cause of severe infestations by OFM in mixed-planting orchards [3,4]. OFM can live on a wide range of hosts and produce many generations annually [5]. Because of the boring damage to tender shoots and fruits, the application of surface pesticides alone cannot achieve the desirable results, and excessive pesticide application can cause many problems, such as environmental pollution, pesticide residue on fruits, loss of natural enemies, and pesticide resistance [6,7].

In recent years, sex-based monitoring as a specific-targeted and eco-friendly alternative to insecticides is used to trap male moths or interfere with their ability to locate females, as well as mating disruption via pheromone attractants [8,9,10,11]. However, only males of OFM are attracted by sex pheromones [12]; males are polygamous and dispersal of mated females is a major contributor to switching between hosts, which increases damage across seasons and crops [1,13]; therefore, the strategy of luring females would be more effective to limit moth populations than sex-pheromone-based techniques. Sugar–acetic acid–ethanol–water mixture (SAEWM) as a food attractant is a pollution-free approach which can be utilized in a variety of traps made with simple and readily accessible materials. These traps have initially proved to perform well on monitoring or trapping insects such as *Coleoptera* and *Diptera* [14,15,16]. Most importantly, because SAEWM is based on the feeding habits of OFM, this technology can trap both female and male adults of OFM and may reduce population of the next generation of OFM [17].

Factors affecting the efficacy of trapping moths by SAEWM include environmental factors such as light intensity, temperature, humidity, and wind direction [18,19,20], as well as nonenvironmental factors such as trap hanging height and space [21,22,23,24]. Currently, studies of SAEWM trapping have primarily focused on the effect of different SAEWM mixing ratios. Few field studies have investigated the effects of the SAEWM trap hanging height on numbers and sex ratio of OFM and numbers of natural enemy insects captured. *Coccinellidae* and *Chrysopidae* are recognized as major natural enemy insects that consume a wide range of insect prey [25,26,27]. There is a concern that if SAEWM-baited traps remove too many natural enemies that the number of other pests in orchards could increase.

In the present study, sex-pheromone and SAEWM-baited traps were hung at different heights to note effects on numbers and sex ratio of adult OFM captured in mixed-planting orchards. The changes in seasonal dynamics, daily flight rhythm, and sex ratio of OFM were observed. Numbers were recorded of natural enemy insects, such as *Coccinellidae* and *Chrysopidae*, captured in SAEWM-baited traps. The aim of this study was to provide a theoretical basis and supporting data for the use of SAEWM-baited traps to monitor and reduce OFM numbers in the field with minimal effect on numbers of natural enemies.

## 2. Results

### 2.1. Monitoring of Adult Moths by SAEWM- and Sex-Pheromone-Baited Traps

#### 2.1.1. Seasonal Flight of OFM

The seasonal changes in numbers of OFM adults captured in SAEWM-baited and sex-pheromone-baited traps in the peach and apple orchards of Tai’an, China, are shown in Figure 1. In these orchards, the total number of moths trapped by the SAEWM-baited traps was less than that by the sex-pheromone-baited traps. However, the trends in the number of moths, as well as the duration time and peak, were basically identical for the SAEWM- and sex-pheromone-baited traps. In 2016, adult moths appeared in the field from early April to early October, and five trapping peaks occurred in the field: 8 April to 14 May, 20 May to 19 June, 22 June to 19 July, 28 July to 25 August, and 28 August to 24 September.

In the peach orchard, moths were trapped by the SAEWM- or sex-pheromone-baited traps primarily during three time periods: 20 May to 19 June, 22 June to 19 July, and 28 July to 25 August (Figure 1a). In the apple orchard, moths were trapped by the SAEWM- or sex-pheromone-baited traps primarily during three time periods: 22 June to 19 July, 28 July to 25 August, and 28 August to 29 September (Figure 1b).

#### 2.1.2. Diurnal Flight of OFM

During the peak flight periods of the first to fourth generations, the diurnal flight of OFM adults in SAEWM-baited traps followed the same trend in both the peach and apple orchards (Figure 2). However, the duration time and peak of adult moth trapping by SAEWM differed during every four continuous 72 h monitoring period. Among the four generations, the daily trapping peak occurred from 18:00 to 20:00, with a small peak on 2–5 July (Figure 2b) and the maximum trapping peak on 1–4 August (Figure 2c), appearing at 4:00–8:00. The number of moths trapped from 4:00 to 8:00 during 3–6 September was similar to that observed from 18:00 to 20:00, during the same period (Figure 2d).

### 2.2. Effect of Trap Hanging Heights on Sex Ratio of OFM Adults Trapped

During the eclosion period of each moth generation in the experimental orchards, adult OFM females were trapped at all the hanging heights at 1–2.5 m above ground, although their numbers differed significantly among the different hanging heights (*p* < 0.05) (Table 1). For both sex-pheromone and SAEWM traps, the total OFM adults were concentrated at 2 and 2.5 m above ground. In the peach orchard, 3675 adult females were trapped by SAEWM in 2016, accounting for 93.27% of all the adult moths trapped by SAEWM that year, whereas 109 adult females were trapped by sex pheromone, accounting for 0.52% of all the adult moths trapped by sex pheromone. In the apple orchard, 2958 adult females were trapped by SAEWM in 2016, accounting for 90.77% of all the adult moths trapped by SAEWM that year, whereas 127 adult females were trapped by sex pheromone, accounting for 0.56% of all the adult moths trapped by sex pheromone that year.

In both the peach and apple orchards, the maximum number of adult females (457.75 and 431.25 per trap) were trapped at 2.5 m above ground by the SAEWM traps, and significantly more than that at the other hanging height (*p* < 0.05). By the SAEWM traps, adult females trapped were concentrated at 2 and 2.5 m above ground. The number of adult females trapped at 2 and 2.5 m above ground accounted for 87.83% and 89.83% of the total number of moths trapped in the peach and apple orchards, respectively. At 1–2.5 m above ground, the proportion of adult females trapped (accounting for all the moths at the same height) generally increased with the hanging height in the apple orchard. However, in the peach orchard, the proportion of females trapped at 2.5 m above ground was slightly lower than that observed at 2 m above ground.

### 2.3. Effects of SAEWM on *Coccinellidae* and *Chrysopidae*

The dynamics of *Coccinellidae* and *Chrysopidae* trapped were demonstrated according to the average number per trap (Figure 3a,b). In 2016, the trapping of *Coccinellidae* by SAEWM-baited traps in Tai’an, China, occurred from 17 May to 10 August and was concentrated between 4 June and 7 July, with 128.7 per trap and 287.2 per trap in the peach and apple orchards, accounting for 98.77% and 99.88% of the total number trapped (130.3 and 287.55 per trap), respectively (Figure 3a). The trapping of *Chrysopidae* by SAEWM occurred from 17 May to 27 September and was concentrated between 1 June and 16 July, with 367.25 per trap and 346.95 per trap in the peach and apple orchards, accounting for 98.84% and 97.73% of the total number trapped (371.55 and 355 per trap), respectively (Figure 3b). As shown in Table 2, both the number of *Coccinellidae* and *Chrysopidae* trapped was concentrated at the heights of 2 m and 2.5 m above ground, significantly greater than that at any other height (*p* < 0.05).

Additionally, the ratio of natural enemies/pests trapped by SAEWM in the experimental mixed-planting orchard was also analyzed. In the present experiment, the number of OFM trapped by SAEWM was 3940 and 3260 in the peach and apple orchards, respectively (Table 1), whereas the total number of *Coccinellidae* and *Chrysopidae* trapped was 10,037 and 12,851 in the peach and apple orchards, respectively (Table 2). Thus, the ratio of natural enemies/pests trapped by SAEWM was 2.5 and 3.9 in the peach and apple orchards, respectively.

## 3. Discussion

### 3.1. Monitoring of OFM by SAEWM

The results of our study showed that SAEWM and sex-pheromone traps are consistent and useful for accurately monitoring the occurrence of moths in the field based on the number of adult moths trapped. Both SAEWM and sex-pheromone traps identified four peaks between late May and late August in the peach orchard and between mid-June and late September in the apple orchard. These peaks may be related to the typical seasonal host relocation of OFM, that is, the larvae of generations 1–2 primarily bore into tender peach shoots, whereas the larvae of generations 3–4 primarily bore into the fruits of peach, pear, and apple trees [4]. Dynamic trends in the diurnal activities of adult moths were similar in the experimental peach and apple orchards. However, the eclosion peak and diurnal rhythm of adult moths were different for each generation. Consequently, the duration time and peak of adult moth trapping were also different, indicating that the feeding time of adult moths was related to the eclosion time of each generation, regardless of the orchard type.

Currently, the field research on diurnal flight in OFM has mainly focused on the mating time, with few studies investigating the feeding-activity rhythms. Gentry et al. observed that OFM mating in the field occurred 2–3 h before sunset [28]. Zhang et al. reported that OFM mating primarily occurs a few hours before the scotophase, which is also an important period of moth oviposition [29]. By trapping males with female pheromones, Zhao et al. observed that the mating of OFM in orchards began after 17:00 and that mating activities peaked from 19:00 to 22:00, with a significant drop-off observed after 20:00 [30]. In our study, the timing of feeding activities was determined via SAEWM trapping. The results were consistent with those obtained by Zhao et al. [30]. Moreover, we also found that feeding and mating activities coexisted from early afternoon to midnight and the number of adult moths trapped increased over time from 4:00 to 8:00. Whether any mating activities occur during this period (4:00–8:00) remains to be further elucidated.

### 3.2. Trapping Effects of SAEWM on OFM

In spite of consistency in monitoring OFM, our investigation into trapping efficacy of SAEWM and sex pheromone confirmed that SAEWM could trap both adult female and male, most of which were adult females. This was in accordance with the previous report that adult female moths showed greater taxis toward SAEWM than adult males due to the feeding requirements of egg-producing females [17]. Furthermore, the number of adult females trapped followed the same pattern as the total number of adult moths. Therefore, compared with sex-pheromone trapping, SAEWM trapping is a more practical method to trap and control OFM. In addition, a few adult females were trapped by the sex-pheromone traps that may have been attracted by hormones released by the males trapped.

From this experiment, adult moths were trapped at heights within 0–2.5 m above ground during the eclosion period, with significantly more moths trapped at 2 and 2.5 m above ground than at any other height. Similarly, Rothschild and Minks [24] observed that more adult moths were captured in traps at 2 m than 1 m or 4 m. Furthermore, our experiments showed that adult moths were primarily active at 2–2.5 m above ground in the orchards, which also revealed that the height range and location had an identical influence on both the feeding and mating activities of OFM. At 0–2.5 m above ground, the number of OFM trapped by sex pheromone was significantly greater than that observed by SAEWM at the same height (*p* < 0.05). However, the difference in trapping efficacies between the SAEWM and sex-pheromone traps decreased as the trap hanging height increased, which was observed in both the peach and apple orchards, indicating that the hanging height of the traps played a key role in OFM trapping, regardless of the orchard type.

### 3.3. Trapping Effects of SAEWM on *Coccinellidae* and *Chrysopidae*

The results of the current study showed that SAEWM can trap *Coccinellidae* and *Chrysopidae* at all heights, and the peak trapping occurred from early June through mid-late July, which is consistent with the results of Wu et al. [31]. The number of *Coccinellidae* and *Chrysopidae* trapped by SAEWM in the peach orchard during the peak trapping period accounted for 98.78% and 97.64% of the total annual number of natural enemies trapped for each group, respectively, whereas accounting for 99.45% and 97.69% in the apple orchard, respectively. The wheat harvest in Tai’an, China, occurs during early June, after which *Coccinellidae* and *Chrysopidae* relocate to surrounding orchards to seek food due to the loss of the wheat fields as a habitat and food resources. Thus, *Coccinellidae* and *Chrysopidae* trapped by SAEWM may have been concentrated during the period after the wheat harvest partly because of their dispersal foraging habits. In the experimental mixed-planting orchard, the number of *Coccinellidae* trapped in the apple orchard by SAEWM was 2.21 times that observed in the peach orchard, whereas the number of *Chrysopidae* trapped was basically the same in both orchards, which released that the composition of the natural enemy population varied with the orchard type.

### 3.4. Effect of SAEWM on Ratio of Natural Enemies/Pests

Based on the efficacy of SAEWM traps for OFM and natural enemies, we also analyzed the influence of SAEWM on the ratio of natural enemies/pests to assess the potential application of this method in the field, and found that the ratio of natural enemies/pests was 2.5 and 3.9 in peach and apple orchards, respectively, suggesting that the utilization of SAEWM during this period would disturb the ecological balance between natural enemies and pests in orchards and, consequently, increase the damage caused by other pests due to lack of natural enemies. Thus, it is advisable to avoid the utilization of SAEWM during the peak trapping period of natural enemies or to replace it with alternative lures, like sex pheromone, that selectively attract and capture only pests. This would conserve numbers of natural enemies and increase the pest-control effects.

## 4. Materials and Methods

### 4.1. Materials

The study was conducted in 2016 within 8-year-old peach and apple mixed-planting orchards adjacent to the Tianpinghu Base of Shandong Institute of Pomology in Tai’an, China (N36°11′57″/E117°2′2″). Each peach and apple orchard was 1.32 ha and measured 110 m from east to west direction by 120 m from north to south. The two orchards were 100 m apart and each had grass planted between the rows. The peach cultivar was Dai Fei trained to Y-shaped trees spaced 1.5 m apart by 4 m between rows. Peaches ripened in mid-August. The apple cultivar was Yanfu No.3 trained to spindle-shaped trees spaced 2 m apart by 4 m between rows. Apples ripened in mid-October. Cultivation and management measures were the same for the peach and apple orchards.

The OFM sex-pheromone dispensers were provided by the Institute of Zoology of the Chinese Academy of Sciences. Each dispenser was a green rubber carrier charged with OFM sex pheromone ((*Z*)-8-dodecenyl acetate:(*E*)-8-dodecenyl acetate:(*Z*)-8-dodecenol (88.5:5.7:2)). The sex-pheromone trap was a basin trap (height of 18 cm and inner diameter of 28–30 cm). The sex-pheromone dispenser was hung at the center of the trap at 3–4 cm above the surface of the trap solution. The SAEWM trap was also a basin trap baited with SAEWM, and both pheromone- and SAEWM-baited traps had a trap basin filled with detergent diluted with water. A trapezoidal frame (Figure 4) was used as a support to adjust the trap height within 0–3 m above ground (patent number: ZL201410693150.5).

The solution in the basin of the sex-pheromone trap was a mixture of detergent and tap water (1:50 ratio of detergent to water). The SAEWM was formulated with white sugar, acetic acid, absolute ethanol, and tap water (3:1:3:80) [6].

### 4.2. Experimental Methods

#### 4.2.1. Trap Deployment

Trapezoidal frames were installed in the east, south, west, and north (20 m apart from the border) of the peach and apple orchards. At each location, 2 frames were installed 20 m apart. The southern frame had sex-pheromone traps, and the northern frame had SAEWM traps. Each trapezoidal frame had a trap hung at each of the five heights (0, 1, 1.5, 2, and 2.5 m), with the liquid surface of each trap aligned with the trap height. The aqueous solutions in both types of traps were replenished as needed to compensate for evaporative losses. The sex-pheromone dispensers and SAEWM were replaced monthly, with spare dispensers stored at 0 to –2 °C.

#### 4.2.2. Seasonal Flight of OFM

The experiment was carried out from 8 April to 12 October 2016. During this period, the number of adult OFM in both traps baited with SAEWM or sex pheromone were recorded every three days. A net was used to filter the adult moths from the traps and transfer them to white plastic bottles filled with an adequate amount of alcohol. Later, the sex of each moth was determined by checking the genitalia under a dissecting microscope, and the numbers were recorded.

#### 4.2.3. Diurnal Flight of OFM

In 2016, real-time monitoring of diurnal flight of OFM began for each of four generations, once the number of adults continued to increase over three consecutive trap counts recorded every three days. Then, moth counts in SAEWM-baited traps at each height were recorded every two hours during the 72 h trapping period from 8:00 on the first day to 8:00 on the fourth day.

#### 4.2.4. Trapping Natural Enemies

The experiment was carried out from 8 April to 12 October 2016, concurrently with the seasonal trapping of OFM. During this period, the numbers of *Coccinellidae* and *Chrysopidae* captured in SAEWM-baited traps at each hanging height were recorded every three days.

#### 4.2.5. Data Analysis

Field monitoring data were collected in 3-day increments and analyzed for variance and significance by Duncan’s new multiple range method with the data processing software SPSS 22.0 (SSPS Inc, Chicago, IL, USA). The results were expressed as the means ± standard error (SE). Graphs were plotted using Excel 2016 (Microsoft Corp, Richmond, WA, USA).

## 5. Conclusions

Compared with sex-pheromone traps, SAEWM traps had precise and potent trapping effects on OFM, especially on adult females. During the eclosion peaks of the first to fourth generations, most of the adult moths were trapped from 4:00 to 8:00 and 18:00 to 20:00 h, compared with other times. Trap hanging height significantly influenced the trapping of OFM, *Coccinellidae*, and *Chrysopidae*. The trapping of *Coccinellidae* and *Chrysopidae* by SAEWM was concentrated from early June through mid-late July and partially overlapped with the peak moth trapping periods (20 May to 19 June and 22 June to 19 July). Therefore, it is advisable to avoid using SAEWM traps during this period or replace them with selective lures that selectively attract and capture only pests and do not trap their natural enemies.

## Figures and Tables

**Figure 1 plants-08-00401-f001:**
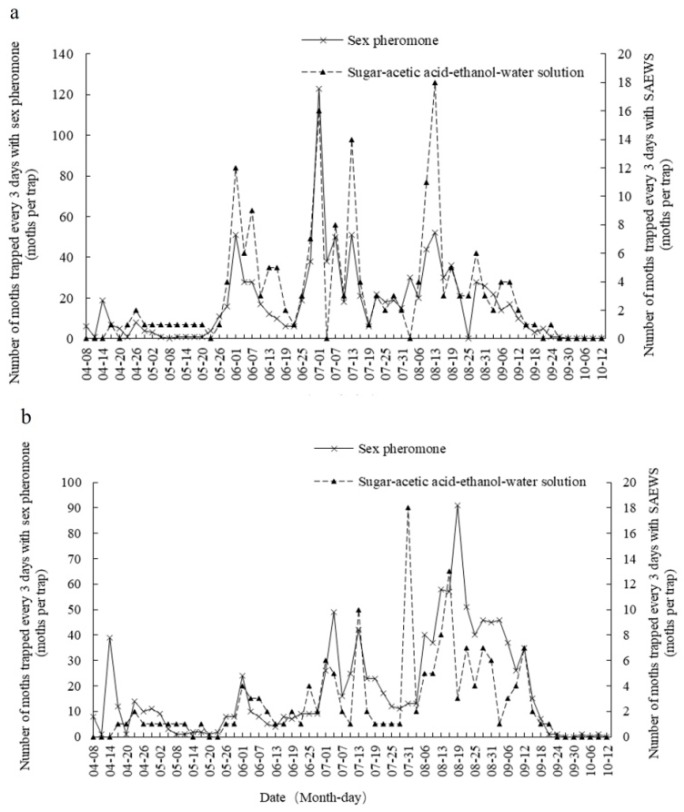
Seasonal changes in flight of *Grapholita molesta* (Busck) adults in the peach (**a**) and apple (**b**) orchards of Tai’an, China, in 2016.

**Figure 2 plants-08-00401-f002:**
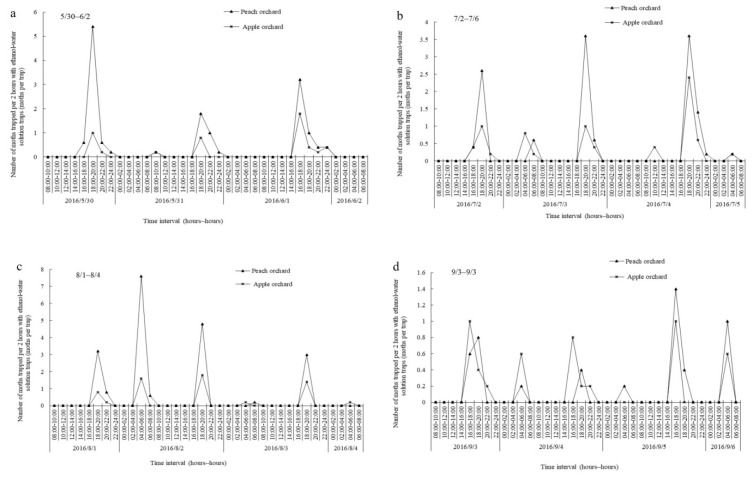
Diurnal flight activity of *Grapholita molesta* (Busck) adults in apple and peach orchards in Tai’an, China, in 2016. (**a**) 30/5–2/6; (**b**) 2/7–6/7; (**c**) 1/8–4/8; and (**d**) 3/9–6/9.

**Figure 3 plants-08-00401-f003:**
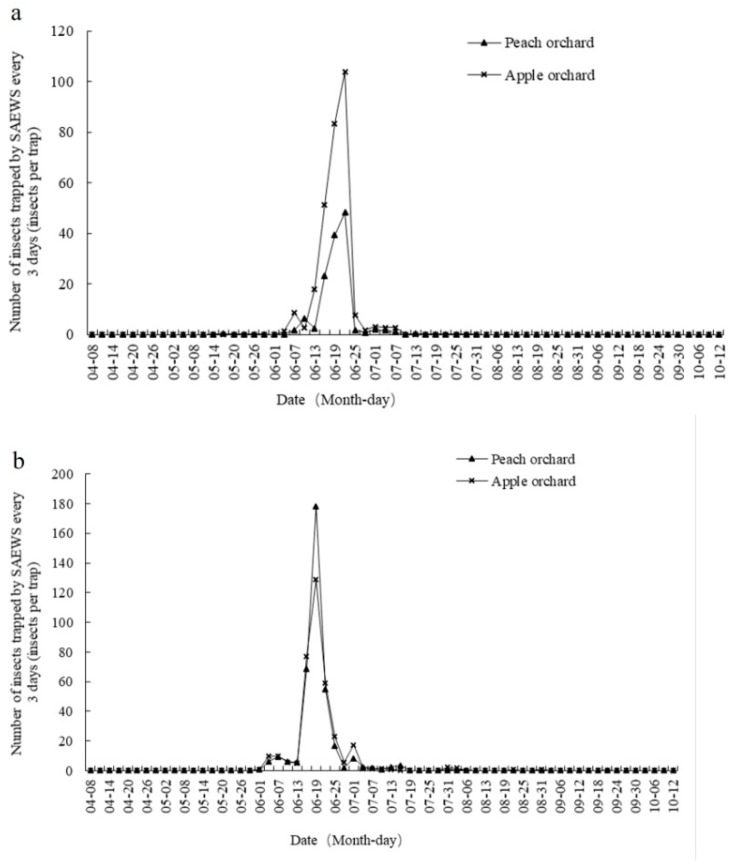
Dynamics of (**a**) *Coccinellidae* and (**b**) *Chrysopidae* trapped by SAEWM-baited traps in Tai’an, China, in 2016.

**Figure 4 plants-08-00401-f004:**
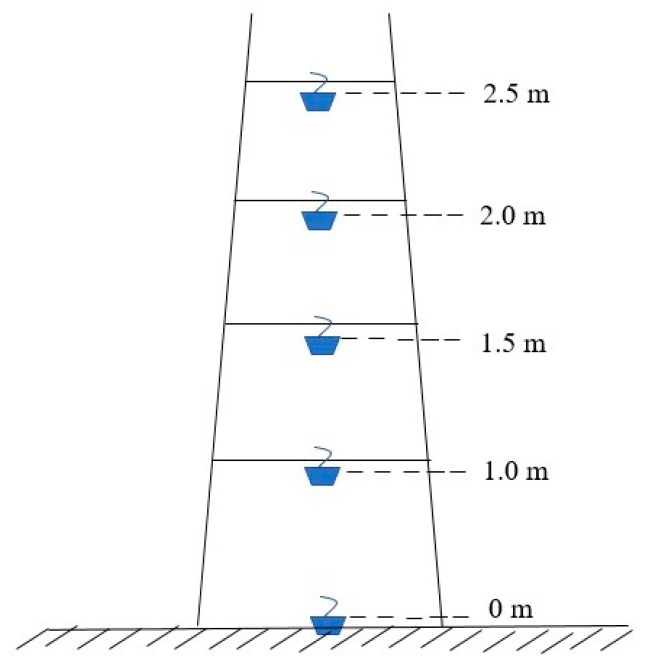
Trapezoidal frame used to hang the traps at different heights.

**Table 1 plants-08-00401-t001:** Trapping efficacy for female/male ratio of OFM adults by sex-pheromone-baited and Sugar–acetic acid–ethanol–water mixture (SAEWM)-baited traps hung at different heights in Tai’an, China, in 2016.

Orchard Type	Trap Hanging Height (m)	Sex-Pheromone Trap			Sugar–Acetic Acid–Ethanol–Water Solutions Trap		
		Number Trapped (moths·trap^–1^)	Percentage of OFM Adults Trapped at Each Height (%)	Percentage of Adult Females Trapped at Each Height (%)	Number Trapped (moths·trap^–1^)	Percentage of OFM Adults Trapped at Each Height (%)	Percentage of Adult Females Trapped at Each Height (%)
Peach orchard	0	33.50 ± 6.03d	0.64	0.00	0.75 ± 0.50d	0.08	100.00
	1	330.00 ± 28.23c	6.27	0.30	12.75 ± 2.50 d	1.29	88.24
	1.5	999.00 ± 80.61b	18.99	0.50	115.75 ± 15.52c	11.75	89.85
	2	1986.75 ± 99.56a	37.77	0.57	366.50 ± 34.28b	37.21	94.13
	2.5	1911.00 ± 229.79a	36.33	0.52	489.25 ± 22.68a	49.67	93.56
Apple orchard	0	62.50 ± 6.86e	1.11	0.00	0.75 ± 0.96d	0.09	100.00
	1	632.50 ± 22.16d	11.18	0.43	12.50 ± 2.08d	1.53	84.00
	1.5	1070.25 ± 150.42c	18.93	0.65	72.75 ± 5.85c	8.93	87.97
	2	1711.50 ± 137.79b	30.27	0.53	259.50 ± 27.23b	31.84	89.88
	2.5	2178.25 ± 223.06a	38.52	0.60	469.50 ± 12.15a	57.61	91.85

Different lowercase (within orchard) letters indicate a significant difference at the 5% level (Duncan’s NMR test).

**Table 2 plants-08-00401-t002:** Effects of trap hanging height on the numbers of *Coccinellidae* and *Chrysopidae* trapped by SAEWM-baited traps in Tai’an, China, in 2016.

Type of Natural Enemy Trapped	Trap Hanging Height (m)	Peach Orchard		Apple Orchard	
		Number Trapped (insects·trap^–1^)	Percentage of Insects Trapped at Each Height (%)	Number of Trapped (insects·trap^–1^)	Percentage of Insects Trapped at Each Height (%)
*Coccinellidae* insects	0	6.00 ± 2.16 d	0.92	39.75 ± 4.03 d	2.76
	1	61.75 ± 3.30 c	9.48	207.75 ± 3.20 c	14.45
	1.5	167.75 ± 18.66 b	25.75	342.75 ± 18.98 b	23.84
	2	238.25 ± 16.72 a	36.57	418.00 ± 18.60 a	29.07
	2.5	177.75 ± 7.93 b	27.28	429.50 ± 16.54 a	29.87
*Chrysopidae* insects	0	5.25 ± 1.89 d	0.28	21.50 ± 3.51 e	1.21
	1	282.75 ± 6.34 c	15.22	360.50 ± 11.03 d	20.31
	1.5	554.00 ± 20.05 a	29.82	403.75 ± 13.15 c	22.75
	2	560.75 ± 19.41 a	30.18	547.25 ± 18.17 a	30.83
	2.5	455.00 ± 22.41 b	24.49	442.00 ± 18.67 b	24.90

Different lowercase (within orchard and insect type) letters indicate a significant difference at the 5% level (Duncan’s NMR test).

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
