# Peer review of "Sugar–Acetic Acid–Ethanol–Water Mixture as a Potent Attractant for Trapping the Oriental Fruit Moth (Lepidoptera: Tortricidae) in Peach–Apple Mixed-Planting Orchards"

_plants, 2019, doi:10.3390/plants8100401_

Round 1

Reviewer 1 Report

The authors have adequately addressed issues raised by the reviewers.

Author Response

Dear reviewer:

We greatly appreciate all the critiques and comments from you. Those comments are extremely helpful for us to improving our paper, and they provide valuable guidance for our future study.

Reviewer 2 Report

Manuscript ID: plants-610211 (original was plants-591815)

Title: Sugar-acetic acid-ethanol-water mixture as a potent attractant for trapping the oriental fruit moth (Lepidoptera: Tortricidae) in peach-apple mixed-planting orchards

Authors: Hao Zhai , Xian-mei Yu , Ya-nan Ma , Yong Zhang , Dan Wang *

Suggestions:

Line 54: change “filed” studies to "field" studies

Line 224: split words “directionby120”   to direction by 120

Line 232: Authors did not do as I suggested – maybe that is not important as I thought

….insert the names and ratios of the OFM sex pheromone components that were added to rubber dispensers.

Line 102: Add details highlighting which height(s) captured significantly more OFM in trap baited with pheromone. You described height differences for SAEWM trap.

Lines 102, 110, 135, 190: The authors still have not done as suggested:

insert in text in parentheses the corresponding values for (F=, df= ; P < or >) whenever they noted a significant or insignificant difference in mean values for that analysis. See lines:

line 102 …differed significantly among the different hanging heights (p<0.05)

line 110 … SAEWM traps, and significantly more than that at the other hanging height.

line 135 …Chrysopidae trapped was concentrated at the heights of 2.0 m and 2.5 m above ground, significantly greater than that at any other height (p<0.05).

line 190 … number of OFM trapped by sex pheromone was significantly greater than that observed by SAEWM at the same height (p<0.05).

Author Response

Dear reviewer:

We greatly appreciate all the critiques and comments from you. Those comments are extremely helpful for us to improving our paper, and they provide valuable guidance for our future study. According to these comments, we have carefully improved our manuscript, and all the revisions are highlighted in red in the text. Please see below point-by-point responses to the reviewers’ comments:

Response to Reviewer 1’s comments

Question 1: 

Line 54: change “filed” studies to "field" studies

Line 224: split words “directionby120”   to direction by 120

R: Thanks for your valuable comment. As suggested, we have carefully corrected the errors you mentioned in the comments.

Page 2, lines 54, “filed” is changed to “field”.

Page 8, lines 225, “directionby120”   is split to "direction by 120".

Question 2: 

Line 232: Authors did not do as I suggested – maybe that is not important as I thought

….insert the names and ratios of the OFM sex pheromone components that were added to rubber dispensers.

R: Thanks for your valuable comment. As suggested, we have added the  components and ratio of sex pheromone (Page 8, lines 233).

Question 3: 

Line 102: Add details highlighting which height(s) captured significantly more OFM in trap baited with pheromone. You described height differences for SAEWM trap.

R: Thanks for your valuable comment. As suggested, we have added the height differences for sex pheromone (Page 4, lines 102~103).

Question 4: 

insert in text in parentheses the corresponding values for (F=, df= ; P < or >) whenever they noted a significant or insignificant difference in mean values for that analysis.

R: Thanks for your valuable comment. As suggested, we have insert "(p<0.05)" in line 102, 112, 136, 191, respectively.

This manuscript is a resubmission of an earlier submission. The following is a list of the peer review reports and author responses from that submission.

Round 1

Reviewer 1 Report

I attached a Word file of the manuscript with "track changes".

Review Comments on September 6, 2019

Journal: Plants (ISSN 2223-7747)

Manuscript ID: plants-591815

Title: Sugar-acetic acid-ethanol-water mixture as a potent attractant for trapping the oriental fruit moth (Lepidoptera: Tortricidae) in peach-apple mixed-planting orchards

Authors: Hao Zhai , Xian-mei Yu , Ya-nan Ma , Yong Zhang , Dan Wang *

There were many long sentences that should be split up.

I had many suggested track changes of the text - see attached Word file of manuscript.

Suggestions:

Line 30: According to ESA Common names, the current correct scientific name is:

Grapholita molesta (Busck)      not Grapholitha

Older literature did use Grapholitha

Line 44, 316: I replaced “migration”  with “dispersal” which is a short-range behavior of moving between adjacent food sources

Line 81: List which OFM sex pheromone components were added to dispensers and ratios of each:

(Z)-8-dodecenyl acetate

(Z)-8-dodecenol

(E)-8-dodecenyl acetate

Line 82: Add citation and reference where this basin trap is first described

Line 90: How much SAEWM solution per trap was used and describe how it was dispensed?

I assume both pheromone and SAEWM baited traps had trap basin filled with water/detergent.

Line 121: We usually use ± standard error (± SE) of the mean and not SD.

Lines 135 and 151: In each figure caption, spell out Grapholita molesta.

Line 151: Add the year 2016 to caption and then remove 2016 from each horizontal date in the x-axis to simplify the x-axis. Since you use hour intervals, I would remove all the zeros in the vertical time values: for example simplify from 08:00-10:00 to   8-10

Lines 155, 172, 182, 193, 201, 235, 239: Whenever you note a significant or insignificant difference in mean values you need to insert in text in parentheses the corresponding (F=, df= ; P < or > values) for that analysis.

Line 165: Change throughout tables: reduce accuracy of mean values to tenths, but can use hundredths for SE values. We usually use standard error of the mean and not SD.

Line 166, 181, 244: In each Table footnote, explain the difference between lower and upper case letters of significance.

For example, did you mean?  

Table 1: Different upper (across periods) and lowercase (within a period) letters indicate a significant difference at the 1 and 5% levels, respectively (Duncan's NMR test).

Table 2: Different upper (across orchards) and lowercase (within orchard) letters indicate a significant difference at the 1 and 5% levels, respectively (Duncan's NMR test).

Table 3: Different upper (across orchards within insect type) and lowercase (within orchard and insect type) letters indicate a significant difference at the 1 and 5% levels, respectively (Duncan's NMR test).

Lines 325-329: Split up sentence. Not sure what you are saying here.

Reviewer 2 Report

If the mass trapping method is the goal in this research, they should mention objectives of research in the Introduction. As the sugar is included in the mixture, authors can refer what is the function? It works as as arrestant when they land on or attach to the source?

I noticed several mistakes as follows:

L.152 Insect name should be italics.

L.292 'hormones' should be 'pheromones', but this speculation is not suitable in this case.

L.354 All the insect scientific names should be italics.

Reviewer 3 Report

This is a nice paper. Understanding the interaction between pest control
and predator abundance contributes, as the authors say, to reduction in
environmental pollution, pesticide residue on fruits, loss of natural
enemies and pesticide resistance. Pest population abundance
determination has potential to make major contribution to reduction in
need for chemical applications while retaining crop protection. The
conclusions which allow advice to be directed to growers regarding
timing is useful. Consideration of effects on predators is good but more
information could be provided on the relevance of Coccinellids and
Chrysopids to OFM control or are they considered important in their
contribution to control of other orchard pests? What are the other
natural enemies OFM?